# ON SCALABLE AND EFFICIENT COMPUTATION OF LARGE SCALE OPTIMAL TRANSPORT

**Yujia Xie, Minshuo Chen, Haoming Jiang, Tuo Zhao, Hongyuan Zha**
Georgia Tech, Atlanta, GA 30332 USA
{xieyujia,mchen393,jianghm}@gatech.edu
tuo.zhao@isye.gatech.edu, zha@cc.gatech.edu

## ABSTRACT

Optimal Transport (OT) naturally arises in many machine learning applications, where we need to handle cross-modality data from multiple sources. Yet the heavy computational burden limits its wide-spread uses. To address the scalability issue, we propose an implicit generative learning-based framework called SPOT (Scalable Push-forward of Optimal Transport). Specifically, we approximate the optimal transport plan by a pushforward of a reference distribution, and cast the optimal transport problem into a minimax problem. We then can solve OT problems efficiently using primal dual stochastic gradient-type algorithms. We also show that we can recover the density of the optimal transport plan using neural ordinary differential equations. Numerical experiments on both synthetic and real datasets illustrate that SPOT is robust and has favorable convergence behavior. SPOT also allows us to efficiently sample from the optimal transport plan, which benefits downstream applications such as domain adaptation.

## 1 INTRODUCTION

The Optimal Transport (OT) problem naturally arises in a variety of machine learning applications, where we need to handle cross-modality data from multiple sources. One example is domain adaptation: We collect multiple datasets from different domains, and we need to learn a model from a source dataset, which can be further adapted to target datasets (Ganin & Lempitsky, 2014; Courty et al., 2017b; Damodaran et al., 2018). Another example is resource allocation: We want to assign a set of assets (one data source) to a set of receivers (another data source) so that an optimal economic benefit is achieved (Santambrogio, 2010; Galichon, 2017). Recent literature has shown that both aforementioned applications can be formulated as optimal transport problems.

The optimal transport problem has a long history, and its earliest literature dates back to Monge (1781). Since then, it has attracted increasing attention and been widely studied in multiple communities such as applied mathematics, probability, economy and geography (Villani, 2008; Carlier, 2012; Gross et al., 2016). Specifically, we consider two sets of data, which are generated from two different distributions denoted by $X \sim \mu$ and $Y \sim \nu$.[1] We aim to find an optimal joint distribution $\gamma$ of $X$ and $Y$, which minimizes the expectation on some ground cost function $c$, i.e.,

$$\gamma^* = \arg\min_{\gamma \in \Pi(\mu,\nu)} \mathbb{E}_{(X,Y)\sim\gamma}[c(X,Y)], \tag{1}$$

The constraint $\gamma \in \Pi(\mu,\nu)$ requires the marginal distribution of $X$ and $Y$ in $\gamma$ to be identical to $\mu$ and $\nu$, respectively. The cost function $c$ measures the discrepancy between input $X$ and $Y$. For cross-modality structured data, the form of $c$ incorporates prior knowledge into optimal transport problem. Existing literature often refers to the optimal expected cost $\mathcal{W}^*(\mu,\nu) = \mathbb{E}_{(X,Y)\sim\gamma^*}[c(X,Y)]$ as *Wasserstein distance* when $c$ is a distance, and $\gamma^*$ as the *optimal transport plan*. For domain adaptation, the function $c$ measures the discrepancy between $X$ and $Y$, and the optimal transport plan $\gamma^*$ essentially reveals the transfer of the knowledge from source $X$ to target $Y$. For resource allocation, the function $c$ is the cost of assigning resource $X$ to receiver $Y$, and the optimal transport plan $\gamma^*$ essentially yields the optimal assignment.

Since equation 1 is an optimization problem over the space of distributions, the problem is infinite dimensional and generally intractable when $\mu$ and $\nu$ are continuous distributions. Therefore, existing literature has resorted to finite dimensional approximations. For example, Cuturi (2013) propose

---

[1]The optimal transport can also handle more than two distributions. See Section 3 for more details.

to discretize the support using a refined grid, and cast equation 1 into a finite dimensional linear programming problem. However, for complex distributions in high dimensions (e.g., images in domain adaptation), the grid size often needs to be exponentially large (e.g., exponential in dimension) to ensure a small approximation error (due to discretization). Under such a regime, conventional linear programming algorithms do not scale well, e.g., the interior point method in conjunction with the Newton's method takes $\mathcal{O}(n^3 \log n)$ time, where $n$ is the grid size. To ease such a scalability issue, Cuturi (2013) propose an entropy regularization-based Sinkhorn algorithm, which requires the computational cost of $\mathcal{O}(n^2)$, but still fail to scale to large problems.

While there exist several scalable stochastic algorithms for computing Wasserstein distance for continuous distributions $\mu$ and $\nu$ (Genevay et al., 2016; Seguy et al., 2017; Yang & Uhler, 2018), they cannot compute the optimal transport plan $\gamma^*$ (see Section 6 for more discussion), which is crucial in the aforementioned applications.

To address the scalability and efficiency issues, we propose a new implicit generative learning-based framework for solving optimal transport problems. Specifically, we approximate $\gamma^*$ by a generative model, which maps from some latent variable $Z$ to $(X, Y)$. For simplicity, we denote

$$\left[ \frac{X}{Y} \right] = G(Z) = \left[ \frac{G_X(Z)}{G_Y(Z)} \right] \quad \text{with} \quad Z \sim \rho, \tag{2}$$

where $\rho$ is some simple latent distribution and $G$ is some operator, e.g., deep neural network or neural ordinary differential equation (ODE). Accordingly, instead of directly estimating the probability density of $\gamma^*$, we estimate the mapping $G$ between $Z$ and $(X, Y)$ by solving

$$G^* = \arg\min_{G} \quad \mathbb{E}_{Z \sim \rho}[c(G_X(Z), G_Y(Z))], \quad \text{subject to} \quad G_X(Z) \sim \mu, \, G_Y(Z) \sim \nu. \tag{3}$$

We then cast equation 3 into a minimax optimization problem using the Lagrangian multiplier method. As the constraints in equation 3 are over the space of continuous distributions, the Lagrangian multiplier is actually infinite dimensional. Thus, we propose to approximate the Lagrangian multiplier by deep neural networks, which eventually delivers a finite dimensional generative learning problem.

Our proposed framework has three major benefits: (1) Our formulated minimax optimization problem can be efficiently solved by primal dual stochastic gradient-type algorithms. Many empirical studies have corroborated that these algorithms can easily scale to very large minimax problems in machine learning (Brock et al., 2018); (2) Our framework can take advantage of recent advances in deep learning. Many empirical evidences have suggested that deep neural networks can effectively adapt to data with intrinsic low dimensional structures (Zhang et al., 2016; Li et al., 2018a). Although they are often overparameterized, due to the inductive biases of the training algorithms, the intrinsic dimensions of deep neural networks are usually controlled very well, which avoids the curse of dimensionality; (3) Our adopted generative models allow us to efficiently sample from the optimal transport plan. This is very convenient for certain downstream applications such as domain adaptation, where we can generate infinitely many data points paired across domains (Liu & Tuzel, 2016).

Moreover, the proposed framework can also recover the density of entropy regularized optimal transport plan. Specifically, we adopt the neural Ordinary Differential Equation (ODE) approach in Chen et al. (2018) to model the dynamics that how $Z$ gradually evolves to $G(Z)$. We then derive the ODE that describes how the density evolves, and solve the density of the transport plan from the ODE. The recovery of density requires no extra parameters, and can be evaluated efficiently.

**Notations**: Given a matrix $A \in \mathbb{R}^{d \times d}$, $\det(A)$ denotes its determinant, $\text{tr}(A) = \sum_i A_{ii}$ denotes its trace, $\|A\|_{\text{F}} = \sqrt{\sum_{i,j} A_{ij}^2}$ denotes its Frobenius norm, and $|A|$ denotes a matrix with $[|A|]_{ij} = |A_{ij}|$. We use $\dim(v)$ to denote the dimension of a vector $v$.

## 2 BACKGROUND

We review some background knowledge on optimal transport and implicit generative learning.

**Optimal Transport**: The idea of optimal transport (OT) originally comes from Monge (1781), which proposes to solve the following problem,

$$T^* = \arg\min_{T(X) \sim \nu} \mathbb{E}_{X \sim \mu}[c(X, T(X))], \tag{4}$$

where $T(\cdot)$ is a mapping from the space of $\mu$ to the space of $\nu$. The optimal mapping $T^*$ is referred to as *Monge map*, and equation 4 is referred to as Monge formulation of optimal transport.

Monge formulation, however, is not necessarily feasible. For example, when $X$ is a constant random variable and $Y$ is not, there does not exist such a map $T$ satisfying $T(X) \sim \nu$. The Kantorovich formulation of our interest in equation 1 is essentially a relaxation of equation 4 by replacing the deterministic mapping with the coupling between $\mu$ and $\nu$. Consequently, *Kantorovich formulation* is guaranteed to be feasible and becomes the classical formulation of optimal transport in existing literature (Benamou et al., 2015; Chizat et al., 2015; Frogner et al., 2015; Solomon et al., 2015).

**Implicit Generative Learning**: For generative learning problems, direct estimation of a probability density function is not always convenient. For example, we may not have enough prior knowledge to specify an appropriate parametric form of the probability density function (pdf). Even when an appropriate parametric pdf is available, computing the maximum likelihood estimator (MLE) can be sometimes neither efficient nor scalable. To address these issues, we resort to implicit generative learning, which do not directly specify the density. Specifically, we consider that the observed variable $X$ is generated by transforming a latent random variable $Z$ (with some known distribution $\rho$) through some unknown mapping $G(\cdot)$, i.e., $X = G(Z)$. We then can train a generative model by estimating $G(\cdot)$ with a properly chosen loss function, which can be easier to compute than MLE. Existing literature usually refer to the distribution of $G(Z)$ as the **push-forward** of reference distribution $\rho$. Such an implicit generative learning approach also enjoys an additional benefit: We only need to choose $\rho$ that is convenient to sample, e.g., uniform or Gaussian distribution, and we then can generate new samples from our learned distribution directly through the estimated mapping $G$ very efficiently.

For many applications, the target distribution can be quite complicated, in contrast to the distribution $\rho$ being simple. This actually requires the mapping $G$ to be flexible. Therefore, we choose to represent G using deep neural networks (DNNs), which are well known for its universal approximation property, i.e., DNNs with sufficiently many neurons and properly chosen activation functions can approximate any continuous functions over compact support up to an arbitrary error. Early empirical evidence, including variational auto-encoder (VAE, Kingma & Welling (2013)) and generative adversarial networks (GAN, Goodfellow et al. (2014)) have shown great success of parameterizing $G$ with DNNs. They further motivate a series of variants, which adopt various DNN architectures to learn more complicated generative models (Radford et al., 2015; Chen et al., 2016; Zhao et al., 2016; Dai et al., 2017; Jiang et al., 2018).

Although the above methods cannot directly estimate the density of the target distribution, for certain applications, we can actually recover the density of $G(Z)$. For example, generative flow methods such as NICE (Dinh et al., 2014), Real NVP (Dinh et al., 2016), and Glow (Kingma & Dhariwal, 2018)) impose sparsity constraints on weight matrices, and exploit the hierarchical nature of DNNs to compute the densities layer by layer. Specifically, NICE proposed in Dinh et al. (2014) denotes the transitions of densities within a neural network as

$$Z \xrightarrow{f_0} h_1 \xrightarrow{f_1} h_2 \cdots h_m \xrightarrow{f_m} G(Z),$$

where $h_i$ represents the hidden units of the $i$-th layer and $f_i$ is the transition function. NICE suggest to restrict the Jacobian matrices of $f_i$'s to be triangular. Therefore, $f_i$'s are reversible and the transition of density in each layer can be easily computed. More recently, Chen et al. (2018) propose a neural ordinary differential equation (neural ODE) approach to compute the transition from $Z$ to $G(Z)$. Specifically, they introduce a dynamical formulation and parameterizing the mapping $G$ using DNNs with recursive structures: They use an ODE to describe how the input $Z$ gradually evolves towards the output $G(Z)$ in continuous time,

$$dz/dt = \xi(z(t), t),$$

where $z(t)$ denotes the continuous time interpolation of $Z$, and $\xi(\cdot, \cdot)$ denotes a feedforward-type DNN. Without loss of generality, we choose $z(0) = Z$ and $z(1) = G(Z)$. Then under certain regularity conditions, the mapping $G(\cdot)$ is guaranteed to be reversible, and the density of $G(Z)$ can be computed in $\mathcal{O}(d)$ time, where $d$ is the dimension of $Z$ (Grathwohl et al., 2018).

## 3 SCALABLE OT WITH PUSHFORWARD

For better efficiency and scalability, we propose a new framework — named SPOT (Scalable Pushforward of Optimal Transport) — for solving the optimal transport problem. Before we proceed with the derivation, we first introduce some notations and assumptions. Recall that

we aim to find an optimal joint distribution $\gamma$ given by equation 1. For simplicity, we assume that the two marginal distributions $X \sim \mu$ and $Y \sim \nu$ have densities $p_X(x)$ and $p_Y(y)$ for $X \in \mathcal{X}$ and $Y \in \mathcal{Y}$ with compact $\mathcal{X}$ and $\mathcal{Y}$, respectively. Moreover, we assume that the joint distribution $\gamma$ has density $p_\gamma$. Then we rewrite equation 1 as the following integral form,

$$p_\gamma^* = \arg\min_{p_\gamma} \int_{x \in \mathcal{X}, y \in \mathcal{Y}} c(x,y) p_\gamma(x,y) dx dy. \quad (5)$$

$$\text{s.t.} \quad \int_{x \in \mathcal{X}} p_\gamma(x,y) dx - p_Y(y) = 0, \quad \forall\, y \in \mathcal{Y}$$

$$\int_{y \in \mathcal{Y}} p_\gamma(x,y) dy - p_X(x) = 0, \quad \forall\, x \in \mathcal{X}$$

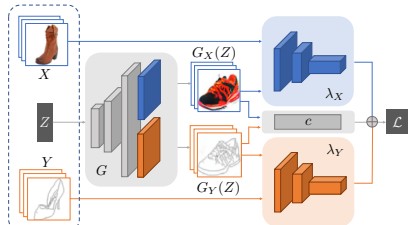

Figure 1: An illustration of SPOT.

We then convert equation 5 into a minmax optimization problem using the Lagrangian multiplier method. Note that equation 5 contains infinitely many constraints, i.e., the equality constraints need to hold for every $x \in \mathcal{X}$ and $y \in \mathcal{Y}$. Therefore, we need infinitely many Lagrangian multipliers. For notational simplicity, we denote the Lagrangian multipliers associated with $x$ and $y$ by two functions $\lambda_X(x) : \mathcal{X} \to \mathbb{R}$ and $\lambda_Y(y) : \mathcal{Y} \to \mathbb{R}$, respectively. Eventually we obtain

$$p_\gamma^* = \arg\min_{p_\gamma} \max_{\lambda_X, \lambda_Y} \int_{x \in \mathcal{X}, y \in \mathcal{Y}} c(x,y) p_\gamma(x,y) dx dy$$

$$+ \int_{y \in \mathcal{Y}} \lambda_Y(y) \left( \int_{x \in \mathcal{X}} p_\gamma(x,y) dx - p_Y(y) \right) dy$$

$$+ \int_{x \in \mathcal{X}} \lambda_X(x) \left( \int_{y \in \mathcal{Y}} p_\gamma(x,y) dy - p_X(x) \right) dx. \quad (6)$$

As mentioned earlier, solving $p_\gamma$ in the space of all continuous distributions is generally intractable. Thus, we adopt the push-forward method, which introduces a mapping $G$ from some latent variable $Z$ to $(X, Y)$. Recall that we denote $(X, Y) = G(Z) = (G_X(Z), G_Y(Z))$ as shown in equation 2. The latent variable $Z$ follows some distribution $\rho$ that is easy to sample. We then rewrite equation 6 as

$$\min_G \max_{\lambda_X, \lambda_Y} \mathbb{E}_{Z \sim \rho}[c(G_X(Z), G_Y(Z))] + \mathbb{E}_{Z \sim \rho}[\lambda_X(G_X(Z))]$$

$$- \mathbb{E}_{X \sim \mu}[\lambda_X(X)] + \mathbb{E}_{Z \sim \rho}[\lambda_Y(G_Y(Z))] - \mathbb{E}_{Y \sim \nu}[\lambda_Y(Y)]. \quad (7)$$

Note that we have replaced the integrals with expectations, since $\int_{x \in \mathcal{X}} p_\gamma(x,y) dx$, $\int_{y \in \mathcal{Y}} p_\gamma(x,y) dy$, $p_X(x)$, and $p_Y(y)$ are probability density functions. Then we further parameterize $G$, $\lambda_X$, and $\lambda_Y$ with neural networks[2]. We denote $\mathcal{G}$ as the class of neural networks for parameterizing $G$ and similarly $\mathcal{F}_X$ and $\mathcal{F}_Y$ as the classes of functions for $\lambda_X$ and $\lambda_Y$, respectively.

Although $\mathcal{G}$, $\mathcal{F}_X$, and $\mathcal{F}_Y$ are finite classes, our parameterization of $G$ cannot exactly represent any continuous distributions of $(X, Y)$ (only up to a small approximation error with sufficiently many neurons). Then the marginal distribution constraints, $G_X(Z) \sim \mu$ and $G_Y(Z) \sim \nu$, are not necessarily satisfied. Therefore, the equilibrium of equation 7 does not necessarily exist, since the Lagrangian multipliers can be unbounded. Motivated by Arjovsky et al. (2017), we require the neural networks for parameterizing $\lambda_X$ and $\lambda_Y$ to be $\eta$-Lipschitz, denoting as $\mathcal{F}_X^\eta$ and $\mathcal{F}_Y^\eta$, respectively. Here $\eta$ can be treated as a tuning parameter, and provides a refined control of the constraint violation. Since each $\eta$-Lipschitz function can be represented by $\eta f$ with $f$ being 1-Lipschitz, we rewrite equation 7 as

$$\min_{G \in \mathcal{G}} \max_{\lambda_X \in \mathcal{F}_X^1, \lambda_Y \in \mathcal{F}_Y^1} \mathbb{E}_{Z \sim \rho}[c(G_X(Z), G_Y(Z))]$$

$$+ \eta \big( \mathbb{E}_{Z \sim \rho}[\lambda_X(G_X(Z))] - \mathbb{E}_{X \sim \mu}[\lambda_X(X)]$$

$$+ \mathbb{E}_{Z \sim \rho}[\lambda_Y(G_Y(Z))] - \mathbb{E}_{Y \sim \nu}[\lambda_Y(Y)] \big). \quad (8)$$

We apply alternating stochastic gradient algorithm to solve equation 8: in each iteration, we perform a few steps of gradient ascent on $\lambda_X$ and $\lambda_Y$, respectively for a fixed $G$, followed by one-step

---

[2]Using a single neural network to parameterize $G$ encourages parameter sharing between $G_X$ and $G_Y$. In fact, we can also parameterize $G_X$ and $G_Y$ with different neural networks.

gradient descent on $G$ for fixed $\lambda_X$ and $\lambda_Y$. We use Spectral Normalization (SN, Miyato et al. (2018)) to control the Lipschitz constant of $\lambda_X$ and $\lambda_Y$ being smaller than 1. Specifically, SN constrains the spectral norm of each weight matrix $W$ by $SN(W) = W/\sigma(W)$ in every iteration, where $\sigma(W)$ denotes the spectral norm of $W$. Note that $\sigma(W)$ can be efficiently approximated by a simple one-step power method (Golub & Van der Vorst, 2001). Therefore, the computationally intensive SVD can be avoided. We summarize the algorithm in Algorithm 1 with SN omitted.

---

**Algorithm 1** Mini-batch Primal Dual Stochastic Gradient Algorithm for SPOT

---

**Require:** Datasets $\{x_i\}_{i=1}^N \sim \mu$, $\{y_j\}_{j=1}^M \sim \nu$; Initialized networks $G$, $\lambda_X$, and $\lambda_Y$ with parameters $w$, $\theta$, and $\beta$, respectively; $\alpha$, the learning rate; $n_{\text{critic}}$, the number of gradient ascent for $\lambda_X$ and $\lambda_Y$; $n$, the batch size

  **while** $w$ not converged **do**

    **for** $t = 1, 2, \cdots, n_{\text{critic}}$ **do**

      Sample mini-batch $\{x_i\}_{i=1}^n$ from $\{x_i\}_{i=1}^N$, $\{y_j\}_{j=1}^n$ from $\{y_j\}_{j=1}^M$, $\{z_k\}_{k=1}^n$ from $\rho$

      $g_\theta \leftarrow \nabla_\theta(\eta \frac{1}{n} \sum_{k=1}^n \lambda_{X,\theta}(G_{X,w}(z_k)) - \eta \frac{1}{n} \sum_{i=1}^n \lambda_{X,\theta}(x_i))$

      $g_\beta \leftarrow \nabla_\beta(\eta \frac{1}{n} \sum_{k=1}^n \lambda_{Y,\beta}(G_{Y,w}(z_k)) - \eta \frac{1}{n} \sum_{i=1}^n \lambda_{Y,\beta}(y_i))$

      $\theta \leftarrow \theta + \alpha g_\theta$, $\beta \leftarrow \beta + \alpha g_\beta$

    **end for**

    Sample mini-batch $\{z_k\}_{k=1}^n$ from $\rho$

    $g_w \leftarrow \nabla_w(\frac{1}{n} \sum_{k=1}^n c(G_{X,w}(z_k), G_{Y,w}(z_k))$

        $+ \eta \frac{1}{n} \sum_{k=1}^n \lambda_{X,\theta}(G_{X,w}(z_k)) + \eta \frac{1}{n} \sum_{k=1}^n \lambda_{Y,\beta}(G_{Y,w}(z_k))$

    $w \leftarrow w + \alpha g_w$

  **end while**

---

**Connection to Wasserstein Generative Adversarial Networks (WGANs)**: Our proposed framework equation 8 can be viewed as a multi-task learning version of Wasserstein GANs (Liu & Tuzel, 2016; Liu et al., 2018). Specifically, the mapping $G$ can be viewed as a *generator* that generates samples in the domains $\mathcal{X}$ and $\mathcal{Y}$. The Lagrangian multipliers $\lambda_X$ and $\lambda_Y$ can be viewed as *discriminators* that evaluate the discrepancies of the generated sample distributions and the target marginal distributions. By restricting $\lambda_X \in \mathcal{F}_X^1$, $\mathbb{E}_{Z\sim\rho}[\lambda_X(G_X(Z))] - \mathbb{E}_{X\sim\mu}[\lambda_X(X)]$ essentially approximates the Wasserstein distance between the distributions of $G_X(Z)$ and $X$ under the Euclidean ground cost (Villani (2008), the same holds for $Y$). Denote

$$\mathcal{R}(G_X, G_Y) = \mathbb{E}_{Z\sim\rho}[c(G_X(Z), G_Y(Z))], \quad \text{and}$$
$$d_w(G_X, X) = \max_{\lambda_X \in \mathcal{F}_X^1} \mathbb{E}_{Z\sim\rho}[\lambda_X(G_X(Z))] - \mathbb{E}_{X\sim\mu}[\lambda_X(X)].$$

Let $d_w(G_Y, Y)$ defined analogously as $d_w(G_X, X)$. We can rewrite equation 8 as

$$\min_{G\in\mathcal{G}} \eta\big(d_w(G_X, X) + d_w(G_Y, Y)\big) + \mathcal{R}(G_X, G_Y), \tag{9}$$

which essentially learns two Wasserstein GANs with a joint generator $G$ through the regularizer $\mathcal{R}$.

**Extension to Multiple Marginal Distributions**: Our proposed framework can be straightforwardly extended to more than two marginal distributions. Consider the ground cost function $c$ taking $m$ inputs $X_1, \ldots, X_m$ with $X_i \sim \mu_i$ for $i = 1, \ldots, m$. Then the optimal transport problem equation 1 becomes the multi-marginal problem (Pass, 2015):

$$\gamma^* = \arg\min_{\gamma\in\Pi(\mu_1,\mu_2,\cdots,\mu_m)} \mathbb{E}_\gamma[c(X_1, X_2, \cdots, X_m)], \tag{10}$$

where $\Pi(\mu_1, \mu_2, \cdots, \mu_m)$ denotes all the joint distributions with marginal distributions satisfying $X_i \sim \mu_i$ for all $i = 1, \ldots, m$. Following the same procedure for two distributions, we cast equation 10 into the following form

$$\min_{G\in\mathcal{G}} \max_{\lambda_{X_i} \in \mathcal{F}_{X_i}^\eta} \mathbb{E}_{Z\sim\rho}[c(G_{X_1}(Z), \cdots, G_{X_m}(Z))]$$
$$+ \sum_{i=1}^m \left(\mathbb{E}_{Z\sim\rho}[\lambda_{X_i}(G_{X_i}(Z))] - \mathbb{E}_{X_i\sim\mu_i}[\lambda_{X_i}(X_i)]\right),$$

where $G$ and $\lambda_{X_i}$'s are all parameterized by neural networks. Existing methods for solving the multi-marginal problem equation 10 suggest to discretize the support of the joint distribution using a refined grid. For complex distributions, the grid size needs to be very large and can be exponential in $m$ (Villani, 2008). Our parameterization method actually only requires at most $2m$ neural networks, which further corroborates the scalability and efficiency of our framework.

## 4 REGULARIZED DENSITY RECOVERY

Existing literature has shown that entropy-regularized optimal transportation outperforms the unregularized counterpart in some applications (Erlander & Stewart, 1990; Cuturi, 2013). This is because the entropy regularizer can tradeoff the estimation bias and variance by controlling the smoothness of the density function.

We demonstrate how to efficiently recover the density $p_\gamma$ of the transport plan with entropy regularization. Instead of parameterizing $G$ by a feedforward neural network, we choose the neural ODE approach, which uses neural networks to approximate the transition from input $Z$ towards output $G(Z)$ in the continuous time. Specifically, we take $z(0) = Z$ and $z(1) = G(Z)$. Let $z(t)$ be the continuous interpolation of $Z$ with density $p(t)$ varying according to time $t$. We split $z(t)$ into $z_1(t)$ and $z_2(t)$ such that $\dim(z_1) = \dim(X)$ and $\dim(z_2) = \dim(Y)$. We then write the neural ODE as

$$dz_1/dt = \xi_1(z(t), t), \quad dz_2/dt = \xi_2(z(t), t), \tag{11}$$

where $\xi_1$ and $\xi_2$ capture the dynamics of $z(t)$. We parameterize $\xi = (\xi_1, \xi_2)$ by a neural network with parameter $w$. We describe the dynamics of the joint density $p(t)$ in the following proposition.

**Proposition 1.** Let $z$, $z_1$, $z_2$, $\xi_1$ and $\xi_2$ be defined as above. Suppose $\xi_1$ and $\xi_2$ are uniformly Lipschitz continuous in $z$ (the Lipschitz constant is independent of $t$) and continuous in $t$. The log joint density satisfies the following ODE:

$$\frac{\partial \log p(t)}{\partial t} = -\left( \mathrm{tr}\left( \frac{\partial \xi_1}{\partial z_1} \right) + \mathrm{tr}\left( \frac{\partial \xi_2}{\partial z_2} \right) \right), \tag{12}$$

where $\frac{\partial \xi_1}{\partial z_1}$ and $\frac{\partial \xi_2}{\partial z_2}$ are Jacobian matrices of $\xi_1$ and $\xi_2$ with respect to $z_1$ and $z_2$, respectively.

Proposition 1 is a direct result of Theorem 1 in Chen et al. (2018). We can now recover the joint density by taking $p_\gamma = p(1)$, which further enables us to efficiently compute the entropy regularizer defined as

$$\mathcal{H}(p_\gamma) = \mathbb{E}_{G(Z) \sim \gamma}[\log p_\gamma(G(Z))].$$

Then we consider the entropy regularized Wasserstein distance $\mathcal{L}_c(G, \lambda_X, \lambda_Y) + \epsilon \mathcal{H}(p_\gamma)$ where $\mathcal{L}_c(G, \lambda_X, \lambda_Y)$ is the objective function in equation 8. Note that here $G$ is a functional operator of $\xi$, and hence parameterized with $w$. The training algorithm follows Algorithm 1, except that updating $G$ becomes more complex due to involving the neural ODE and the entropy regularizer.

To update $G$, we are essentially updating $w$ using the gradient $g_w = \partial(\mathcal{L}_c + \epsilon \mathcal{H})/\partial w$, where $\epsilon$ is the regularization coefficient. First we compute $\partial \mathcal{L}_c/\partial w$. We adopt the integral form from Chen et al. (2018) in the following

$$\frac{\partial \mathcal{L}_c}{\partial w} = -\int_0^1 a(t)^\top \frac{\partial \xi(z(t), t)}{\partial w} dt, \tag{13}$$

where $a(t) = \partial \mathcal{L}_c/\partial z(t)$ is the so-called "adjoint variable". The detailed derivation is slightly involved due to the complicated terms in the chain rule. We refer the readers to Chen et al. (2018) for a complete argument. The advantage of introducing $a(t)$ is that we can compute $a(t)$ using the following ODE,

$$\frac{da(t)}{dt} = -a(t)^\top \frac{\partial \xi(z(t), t)}{\partial z}.$$

Then we can use a well developed numerical method to compute equation 13 efficiently (Davis & Rabinowitz, 2007). Next, we compute $\partial \mathcal{H}/\partial w$ in a similar procedure with $a(t)$ replaced by $b(t) = \partial \mathcal{H}/\partial \log p(t)$. We then write

$$\frac{\partial \mathcal{H}}{\partial w} = -\int_0^1 b(t)^\top \frac{\partial \log p(t)}{\partial w} dt.$$

Using the same numerical method, we can compute $\partial \mathcal{H}/\partial w$, which eventually allows us to compute $g_w$ and update $w$.

## 5 EXPERIMENTS

We evaluate the SPOT framework on various tasks: Wasserstein distance approximation, density recovery, paired sample generation and domain adaptation. All experiments are implemented with PyTorch using one GTX1080Ti GPU and a Linux desktop computer with 32GB memory, and we adopt the Adam optimizer with configuration parameters 0.5 and 0.999 (Kingma & Ba, 2014).

## 5.1 WASSERSTEIN DISTANCE (WD) APPROXIMATION

We first demonstrate that SPOT can accurately and efficiently approximate the Wasserstein distance. We take the Euclidean ground cost, i.e. $c(x, y) = \|x - y\|$. Then $\mathbb{E}_{G(Z) \sim \gamma^*}[c(G_X(Z), G_Y(Z))]$ essentially approximates the Wasserstein distance. We take the marginal distributions $\mu$ and $\nu$ as two Gaussian distributions in $\mathbb{R}^2$ with the same identity covariance matrix. The means are $(-2.5, 0)^\top$ and $(2.5, 0)^\top$, respectively. We find the Wasserstein distance between $\mu$ and $\nu$ equal to 5 by evaluating its closed-form solution. We generate $n = 10^5$ samples from both distributions $\mu$ and $\nu$, respectively. Note that naively applying discretization-based algorithms by dividing the support according to samples requires at least 40 GB memory, which is beyond the memory capability.

We parameterize $G_X$, $G_Y$, $\lambda_X$, and $\lambda_Y$ with fully connected neural networks without sharing parameters. All the networks use the Leaky-ReLU activation Maas et al. (2013). $G_X$ and $G_Y$ have 2 hidden layers. $\lambda_X$ and $\lambda_Y$ have 1 hidden layer. The latent variable $Z$ follows the standard Gaussian distribution in $\mathbb{R}^2$. We take the batch size equal to 100.

**WD vs. Number of Epochs.** We compare the algorithmic behavior of SPOT and Regularized Optimal Transport (ROT, Seguy et al. (2017)) with different regularization coefficients. For SPOT, we set the number of units in each hidden layer equal to 8 and $\eta = 10^4$. For ROT, we adopt the code from the authors[3] with only different input samples, learning rates, and regularization coefficients.

Figure 2 shows the convergence behavior of SPOT and ROT for approximating the Wasserstein distance between $\mu$ and $\nu$ with different learning rates. We observe that SPOT converges to the true Wasserstein distance with only 0.6%, 0.3%, and 0.3% relative errors corresponding to Learning Rates (LR) $10^{-3}$, $10^{-4}$, and $10^{-5}$, respectively. In contrast, ROT is very sensitive to its regularization coefficient. Thus, it requires extensive tuning to achieve a good performance.

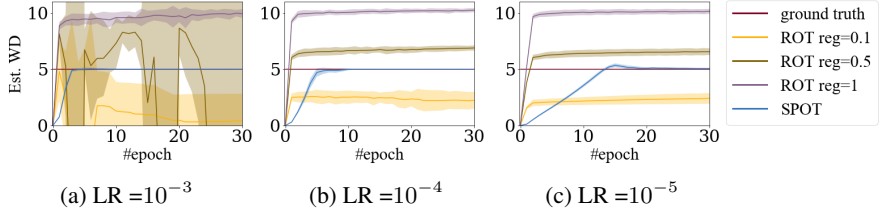

| (a) LR =$10^{-3}$ | (b) LR =$10^{-4}$ | (c) LR =$10^{-5}$ |

Figure 2: Comparison of convergence between SPOT and ROT. All the curves are averaged over 50 runs with different random seeds, and the shaded areas represent the standard deviation.

**WD vs. Number of Hidden Units.** We then explore the adaptivity of SPOT by increasing the network size, while the input data are generated from some low dimensional distribution. Specifically, the number of hidden units per layer varies from 2 to $2^{10}$. Recall that we parameterize $G$ with two 2-hidden-layer neural networks, and $\lambda_X$, $\lambda_Y$ with two 1-hidden-layer neural networks. Accordingly, the number of parameters in $G$ varies from 36 to about $2 \times 10^6$, and that in $\lambda_X$ or $\lambda_Y$ varies from 12 to about 2,000. The tuning parameter $\eta$ also varies corresponding to the number of hidden units in $\lambda_X$, $\lambda_Y$. We use $\eta = 10^5$

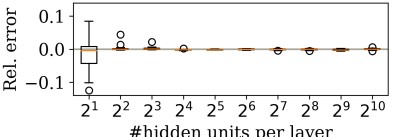

Figure 3: Box plots of relative errors of the estimated Wasserstein distance with respect to the number of hidden units per layer. The results are averaged over 50 independent runs.

for $2^1, 2^2$ and $2^3$ hidden units per layer, $\eta = 2 \times 10^4$ for $2^4, 2^5$ and $2^6$ hidden units per layer, $\eta = 10^4$ for $2^7$ and $2^8$ hidden units per layer, $\eta = 2 \times 10^3$ for $2^9$, and $2^{10}$ hidden units per layer.

Figure 3 shows the estimated WD with respect to the number of hidden units per layer. For large neural networks that have $2^9$ or $2^{10}$ hidden units per layer, i.e., $5.2 \times 10^5$ or $2.0 \times 10^6$ parameters, the number of parameters is far larger than the number of samples. Therefore, the model is heavily overparameterized. As we can observe in Figure 3, the relative error however, does not increase as the number of parameters grows. This suggests that SPOT is robust with respect to the network size.

## 5.2 DENSITY RECOVERY

We demonstrate that SPOT can effectively recover the joint density with entropy regularization. We adopt the neural ODE approach as described in Section 4. Denote $\phi(a, b)$ as the density

---

[3]https://github.com/vivienseguy/Large-Scale-OT

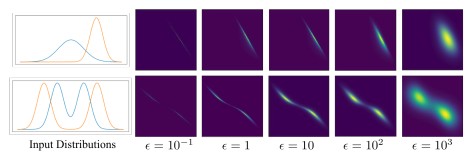

Figure 5: Generated samples of SPOT and CoGAN on the MNIST-MNISTM task.

of the Gaussian distribution $N(a, b)$. We take the marginal distributions $\mu$ and $\nu$ as (1) Gaussian distributions $\phi(0, 1)$ and $\phi(2, 0.5)$; (2) mixtures of Gaussian $\frac{1}{2}\phi(-1, 0.5) + \frac{1}{2}\phi(1, 0.5)$ and $\frac{1}{2}\phi(-2, 0.5) + \frac{1}{2}\phi(2, 0.5)$. The ground cost is the Euclidean square function, i.e., $c(x, y) = \|x - y\|^2$. We run the training algorithm for $6 \times 10^5$ iterations and in each iteration, we generate 500 samples from $\mu$ and $\nu$, respectively. We parameterize $\xi$ with a 3-hidden-layer fully-connected neural network with 64 hidden units per layer, and the latent dimension is 2. We take $\eta = 10^6$.

Figure 4 shows the input marginal densities and heat maps of output joint densities. We can see that a larger regularization coefficient $\epsilon$ yields a smoother joint density for the optimal transport plan. Note that with continuous marginal distributions and the Euclidean square ground cost, the joint density of the unregularized optimal transport degenerates to a generalized impulse function (i.e., a generalized Dirac $\delta$ function that has nonzero value on a manifold instead of one atom, as shown in Rachev (1985); Onural (2006)). Entropy regularization prevents such degeneracy by enforcing smoothness of the density.

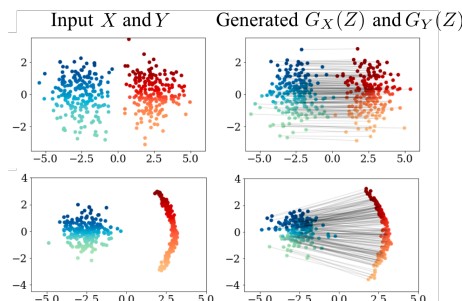

Figure 4: Visualization of the marginal distributions and the joint density of the optimal transport plan.

### 5.3 SAMPLE GENERATION

We show that SPOT can generate paired samples $(G_X(Z), G_Y(Z))$ from unpaired data $X$ and $Y$ that are sampled from marginal distributions $\mu$ and $\nu$, respectively.

**Synthetic Data.** We take the squared Euclidean cost, i.e. $c(x, y) = \|x - y\|^2$, and adopt the same implementation and sample size as in Section 5.1 with learning rate $10^{-3}$ and 32 hidden units per layer. Figure 6 illustrates the input samples and the generated samples with two sets of different marginal distributions: The upper row corresponds to the same Gaussian distributions as in Section 5.1. The lower row takes $X$ as Gaussian distribution with mean $(-2.5, 0)^\top$ and covariance $0.5I$, $Y$ as $(\sin(Y_1) + Y_2, 2Y_1 - 3)^\top$, where $Y_1$ follows a uniform distribution on $[0, 3]$, and $Y_2$ follows a Gaussian distribution $N(2, 0.1)$. We observe that the generated samples and the input samples are approximately identically distributed. Additionally, the paired relationship is as expected – the upper mass is transported to the upper region, and the lower mass is transported to the lower region.

Figure 6: Visualization of input samples and generated samples. The black lines represent the paired relation.

**Real Data.** We next show SPOT is able to generate high quality paired samples from two unpaired real datasets: MNIST (LeCun et al., 1998) and MNISTM (Ganin & Lempitsky, 2014). The handwritten digits in MNIST and MNISTM datasets have different backgrounds and foregrounds (see Figrue 5). The digits in paired images however, are expected to have similar contours. We leverage this prior knowledge[4] by adopting a semantic-aware cost function (Li et al., 2018b) to extract the edge of handwritten letters, i.e., we use the following cost function

$$c(x, y) = \sum_{i=1}^{2} \sum_{j=1}^{3} \||C_i * x_j| - |C_i * y_j|\|_{\mathrm{F}},$$

where $C_1$ and $C_2$ denote the Sobel filter (Sobel, 1990), and $x_j$'s and $y_j$'s are the three channels of RGB images. The operator $*$ denotes the matrix convolution. We set

$$C_1 = \begin{bmatrix} -1 & 0 & 1 \\ -2 & 0 & 2 \\ -1 & 0 & 1 \end{bmatrix} \text{ and } C_2 = \begin{bmatrix} 1 & 2 & 1 \\ 0 & 0 & 0 \\ -1 & -2 & -1 \end{bmatrix},$$

with $C_1$ and $C_2$ defining two extraction directions.

---

[4]For OT problems, $c$ can be viewed as a way to add prior knowledge to the problem (Peyré et al., 2017).

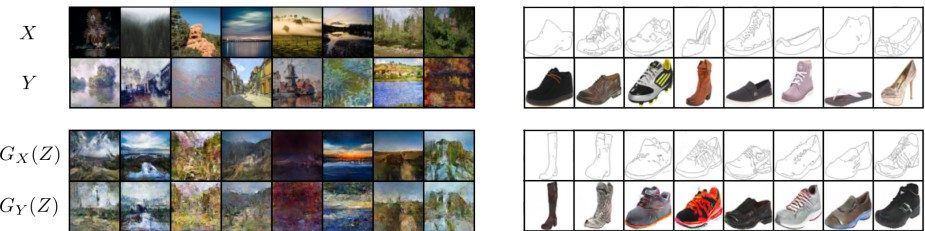

Figure 7: Generated samples of SPOT on Photos-Monet and Sketches-Shoes datasets.

We now use separate neural networks to parameterize $G_X$ and $G_Y$ instead of taking $G_X$ and $G_Y$ as outputs of a common network. Note that $G_X$ and $G_Y$ does not share parameters. Specifically, we use two 4-layer convolutional layers in each neural network for $G_X$ or $G_Y$, and two 5-layer convolutional neural networks for $\lambda_X$ and $\lambda_Y$. More detailed network settings are provided in Appendix A.2. The batch size is 32, and we train the framework with $2 \times 10^5$ iterations until the generated samples become stable.

Figure 5 shows the generated samples of SPOT. We also reproduce the results of CoGAN with the code from the authors[5]. As can be seen, with approximately the same network size, SPOT yields paired images with better quality than CoGAN: The contours of the paired results of SPOT are nearly identical, while the results of CoGAN have no clear paired relation. Besides, the images corresponding to $G_Y(Z)$ in SPOT have colorful foreground and background, while in CoGAN there are only few colors. Recall that in SPOT, the paired relation is encouraged by ground cost $c$, and in CoGAN it is encouraged by sharing parameters. By leveraging prior knowledge in ground cost $c$, the paired relation is more accurately controlled without compromising the quality of the generated images.

We further test our framework on more complex real datasets: Photo-Monet dataset Zhu et al. (2017) and Edges-Shoes dataset Isola et al. (2017). We adopt the Euclidean cost function for Photo-Monet dataset, and the semantic-aware cost function as in MNIST-MNISTM for Edges-Shoes dataset. Other implementations remain the same as the MNIST-MINSTM experiment.

Figure 7 demonstrates the generated samples of both datasets. We observe that the generated images have a desired paired relation: For each $Z$, $G_X(Z)$ and $G_Y(Z)$ gives a pair of corresponding scenery and shoe. The generated images are also of high quality, especially considering that Photo-Monet dataset is a pretty small but complex dataset with 6,288 photos and 1,073 paintings.

## 5.4 DOMAIN ADAPTATION

Optimal transport has been used in domain adaptation, but existing methods are either computationally inefficient (Courty et al., 2017a; Damodaran et al., 2018), or cannot achieve a state-of-the-art performance (Seguy et al., 2018). Here, we demonstrate that SPOT can tackle large scale domain adaptation problems with state-of-the-art performance.

In particular, we receive labeled source data $\{x_i\} \sim \mu$, where each data point is associated with a label $v_i$, and target data $\{y_j\} \sim \nu$ with unknown labels. For simplicity, we use $X$ and $Y$ to denote the random vectors following distributions $\mu$ and $\nu$, respectively. The two distributions $\mu$ and $\nu$ can be coupled in a way that each paired samples of $(X, Y)$ from the coupled joint distribution are likely to have the same label. In order to identify such coupling information between source and target data, we propose a new OT-based domain adaptation method — DASPOT (Domain Adaptation with SPOT) as follows.

Specifically, we jointly train an optimal transport plan and two classifiers for $X$ and $Y$ (denoted by $D_X$ and $D_Y$, respectively). Each classifier is a composition of two neural networks — an embedding network and a decision network. For simplicity, we denote $D_X = D_{e,X} \circ D_{c,X}$, where $D_{e,X}$ denotes the embedding network, and $D_{c,X}$ denotes the decision network (respectively for $D_Y = D_{e,Y} \circ D_{c,Y}$). We expect the embedding networks to extract high level features of the source and target data, and then find an optimal transport plan to align $X$ and $Y$ based on these high level features using SPOT. Here we choose a ground cost $c(x, y) = \|D_{e,X}(x) - D_{e,Y}(y)\|^2$. Let $G$ denote the generator of SPOT. The Wasserstein distance of such an OT problem can be written as $\mathbb{E}_Z \|D_{e,X}(G_X(Z)) - D_{e,Y}(G_Y(Z))\|^2$.

---

[5]https://github.com/mingyuliutw/CoGAN

Meanwhile, we train $D_X$ by minimizing the empirical risk $\frac{1}{n}\sum_{i=1}^{n}[\mathcal{E}(D_X(x_i), v_i)]$, where $\mathcal{E}$ denotes the cross entropy loss function, and train $D_Y$ by minimizing

$$\mathbb{E}_Z[\mathcal{E}(D_Y(G_Y(Z)), \arg\max_k[D_X(G_X(Z))]_k)], \tag{14}$$

where $[v]_k$ denotes the $k$-th entry of the vector $v$. The risk function defined in equation 15 essentially encourages $D_X$ and $D_Y$ to predict each paired (synthetic) samples of $(G_X(Z), G_Y(Z))$ to have the same label.

Eventually, the joint training optimize

$$\min_{D_X, D_Y, G} \max_{\lambda_X, \lambda_Y} \mathcal{L}_c(G, \lambda_X, \lambda_Y) + \frac{\eta_s}{n}\sum_{i=1}^{n}[\mathcal{E}(D_X(x_i), v_i)]$$
$$+\eta_{da}\mathbb{E}_Z[\mathcal{E}(D_Y(G_Y(Z)), \arg\max_k[D_X(G_X(Z))]_k)],$$

where $\mathcal{L}_c(G, \lambda_X, \lambda_Y)$ is the objective function of OT problem in equation 8 with $c$ defined above, and $\eta_s, \eta_{da}$ are the tuning parameters. We choose $\eta_s = 10^3$ for all experiments. We set $\eta_{da} = 0$ for the first $10^5$ iteration to wait the generators to be well trained. Then we set $\eta_{da} = 10$ for the next $3 \times 10^5$ iteration. We take totally $4 \times 10^5$ iterations, and set the learning rate equal to $10^{-4}$ and batch size equal to $128$ for all experiments.

We evaluate DASPOT with the MNIST, MNISTM, USPS (Hull, 1994), and SVHN (Netzer et al., 2011) datasets. We denote a domain adaptation task as Source Domain → Target Domain. For the tasks MNIST → USPS, USPS → MNIST and MNIST → MNISTM, we use three 4-layer networks for $D, \lambda_X$, and $\lambda_Y$, and two 5-layer networks for $G_X$ and $G_Y$. For the task SVHN → MNIST, we use three 5-layer downsampling ResNets He et al. (2016) for $D, \lambda_X$, and $\lambda_Y$, and two 5-layer up-sampling ResNets for $G_X$ and $G_Y$. More detailed implementations are provided in Appendices A.2 and A.3.

We compare the performance of DASPOT with other optimal transport based domain adaptation methods: ROT (Seguy et al., 2018), StochJDOT (Damodaran et al., 2018) and DeepJDOT (Damodaran et al., 2018). As can be seen in Table 1, DASPOT achieves equal or better performances on all the tasks.

Table 1: *Domain Adaptation Experiments on multiple tasks.*

| Source | MNIST | USPS | SVHN | MNIST |
|--------|-------|------|------|-------|
| Target | USPS | MNIST | MNIST | MNISTM |
| ROT | 72.6% | 60.5% | 62.9% | − |
| StochJDOT | 93.6% | 90.5% | 67.6% | 66.7% |
| DeepJDOT | 95.7% | 96.4% | **96.7%** | 92.4% |
| DASPOT | **97.5%** | **96.5%** | 96.2% | **94.9%** |

Moreover, we show that DeepJDOT is not as efficient as DASPOT. For example, in the MNIST → USPS task, DASPOT requires 169s running time to achieve a $95\%$ accuracy, while DeepJDOT requires 518s running time to achieve the same accuracy. The reason behind is that DeepJDOT needs to solve a series of optimal transport problems using Sinkhorn algorithm. The implementation of DeepJDOT is adapted from the authors' code[6].

## 6 DISCUSSION

Existing literature shows that several stochastic algorithms can efficiently compute the Wasserstein distance between two continuous distributions. These algorithms, however, only apply to the dual of the OT problem equation 1, and cannot provide the optimal transport plan. For example, Genevay et al. (2016) suggest to expand the dual variables in two reproducing kernel Hilbert spaces. They then apply the Stochastic Averaged Gradient (SAG) algorithm to compute the optimal objective value of OT with continuous marginal distributions or semi-discrete marginal distributions (i.e., one marginal distribution is continuous and the other is discrete). The follow-up work, Seguy et al. (2017), parameterize the dual variables with neural networks and apply the Stochastic Gradient Descent (SGD) algorithm to eventually achieve a better convergence. These two methods can only provide the optimal transport plan and recover the joint density when the densities of the marginal distributions are known. This is prohibitive in most applications, since we only have access to the empirical data. Our framework actually allows us to efficiently compute the joint density from the transformation of the latent variable $Z$ as in Section 4.

---

[6]https://github.com/bbdamodaran/deepJDOT

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

# Appendix

## A   NETWORK ARCHITECTURE

### A.1   NO-SHARING NETWORK

The CNN architecture for experiments in Section 5.3. Table 2 shows the architecture of two mappings $G_X$ and $G_Y$. The two mappings have identical architechture.

Table 2: The CNN architecture for experiments of real datasets in Section 5.3.

| **Input**: | $z \in \mathbb{R}^{100} \sim \mathcal{N}(0, I)$ | |
|---|---|---|
| | Convolution Filter | Activation |
| Deconv: | $[4 \times 4, 512,$ stride = 1, padding=0] | BN, ReLU |
| Deconv: | $[4 \times 4, 256,$ stride = 2, padding=1] | BN, ReLU |
| Deconv: | $[4 \times 4, 128,$ stride = 2, padding=1] | BN, ReLU |
| Deconv: | $[4 \times 4, 64,$  stride = 2, padding=1] | BN, ReLU |
| Deconv: | $[4 \times 4, 3,$   stride = 2, padding=1] | Tanh |

Table 3 shows the architecture of two discriminators $\lambda_X, \lambda_Y$. The two networks have identical architechture and do not share parameters.

Table 3: The CNN architecture of $\lambda_X, \lambda_Y$ for experiments of real datasets in Section 5.3.

| **Input**: | Image $x \in \mathbb{R}^{64 \times 64 \times 3} \sim \mu$ or $\nu$ | |
|---|---|---|
| | Convolution Filter | Activation |
| Conv: | $[4 \times 4, 64,$   stride = 1, padding=0] | ReLU |
| Conv: | $[4 \times 4, 128,$ stride = 2, padding=1] | BN, ReLU |
| Conv: | $[4 \times 4, 256,$ stride = 2, padding=1] | BN, ReLU |
| Conv: | $[4 \times 4, 512,$ stride = 2, padding=1] | BN, ReLU |
| Conv: | $[4 \times 4, 1,$    stride = 1, padding=0] | – |

### A.2   CONVOLUTIONAL NETWORK

The CNN architecture for USPS, MNIST and MNISTM. PReLU activation is applied He et al. (2015). Table 4 shows the architecture of two generators $G_X$ and $G_Y$. The last column in Table 4 means whether $G_X$ and $G_Y$ share the same parameter.

Table 4: The CNN generater architecture for USPS, MNIST and MNISTM. $ch = 1$ for USPS and MNIST; $ch = 3$ for MNISTM.

| **Input**: | $z \in \mathbb{R}^{100} \sim \mathcal{N}(0, I)$ | | |
|---|---|---|---|
| | Convolution Filter | Activation | Shared |
| Deconv: | $[4 \times 4, 1024,$ stride = 1, padding=0] | BN, PReLU | True |
| Deconv: | $[3 \times 3, 512,$  stride = 2, padding=1] | BN, PReLU | True |
| Deconv: | $[3 \times 3, 256,$  stride = 2, padding=1] | BN, PReLU | True |
| Deconv: | $[3 \times 3, 128,$  stride = 2, padding=1] | BN, PReLU | True |
| Deconv: | $[3 \times 6, ch,$   stride = 1, padding=1] | Sigmoid | False |

Table 5 shows the architecture of two discriminators $\lambda_X, \lambda_Y$, and two classifiers $D_X, D_Y$. The last column in Table 4 uses $(\cdot, \cdot)$ to denote which group of discriminators share the same parameter.

Table 5: The CNN discriminator architecture for USPS, MNIST and MNISTM. $ch = 1$ for USPS and MNIST; $ch = 3$ for MNISTM. $ch_o = 1$ for $\lambda_X$ and $\lambda_Y$; $ch_o = 10$ for $D_X$ and $D_Y$.

| **Input**: | Image $x \in \mathbb{R}^{28 \times 28 \times ch} \sim \mu$ or $\nu$ | | |
|---|---|---|---|
| | Convolution Filter | Activation | Shared |
| Conv: | $[5 \times 5, 20,\quad$ stride = 1, padding=0] | MaxPooling(2,2) | $(\lambda_X, D_X);(\lambda_Y, D_Y)$ |
| Conv: | $[5 \times 5, 50,\quad$ stride = 1, padding=0] | MaxPooling(2,2) | $(\lambda_X, \lambda_Y, D_X, D_Y)$ |
| Conv: | $[4 \times 4, 500,$ stride = 1, padding=0] | PReLU | $(\lambda_X, \lambda_Y, D_X, D_Y)$ |
| Conv: | $[1 \times 1, ch_o,$ stride = 1, padding=0] | – | $(\lambda_X); (\lambda_Y); (D_X, D_Y)$ |

## A.3 RESIDUAL NETWORK

The ResNet architecture for SVHN $\rightarrow$ MNIST. Table 6 shows the architecture of two generators $G_X$ and $G_Y$. The last column in Table 6 means whether $G_X$ and $G_Y$ share the same parameter. The Residual block is the same as the one in Miyato et al. (2018).

Table 6: The ResNet generater architecture for SVHN $\rightarrow$ MNIST. $ch = 1$ for MNIST; $ch = 3$ for SVHN.

| **Input**: | $z \in \mathbb{R}^{100} \sim \mathcal{N}(0, I)$ | | |
|---|---|---|---|
| | Layer Size | Activation | Shared |
| Linear: | $100 \rightarrow 4 \times 4 \times 128$ | – | True |
| ResBlocks: | [128, Up-sampling] | – | True |
| ResBlocks: | [128, Up-sampling] | – | True |
| ResBlocks: | [128, Up-sampling] | BN,PReLU | True |
| Conv: | $[3 \times 3, ch,$ stride = 1, padding =0] | Sigmoid | False |

Table 7 shows the architecture of two discriminators $\lambda_X, \lambda_Y$, and two classifiers $D_X, D_Y$. The last column in Table 7 uses $(\cdot, \cdot)$ to denote which group of discriminators share the same parameter.

Table 7: The ResNet discriminator architecture for SVHN $\rightarrow$ MNIST. $ch = 1$ for MNIST; $ch = 3$ for SVHN. $ch_o = 1$ for $\lambda_X$ and $\lambda_Y$; $ch_o = 10$ for $D_X$ and $D_Y$.

| **Input**: | Image $x \in \mathbb{R}^{28 \times 28 \times ch} \sim \mu$ or $\nu$ | | |
|---|---|---|---|
| | Layer Size | Activation | Shared |
| ResBlocks: | [128, Down-Sampling] | – | $(\lambda_X, D_X);(\lambda_Y, D_Y)$ |
| ResBlocks: | [128, Down-Sampling] | – | $(\lambda_X, \lambda_Y, D_X, D_Y)$ |
| ResBlocks: | [128, Down-Sampling] | – | $(\lambda_X, \lambda_Y, D_X, D_Y)$ |
| Conv: | $[4 \times 4, 500,$ stride = 1, padding=0] | PReLU | $(\lambda_X, \lambda_Y, D_X, D_Y)$ |
| Conv: | $[1 \times 1, ch_o,$ stride = 1, padding=0] | – | $(\lambda_X); (\lambda_Y); (D_X, D_Y)$ |

