# OpenReview forum: "On Scalable and Efficient Computation of Large Scale Optimal Transport"
_ICLR.cc/2019/Workshop/DeepGenStruct — DeepGenStruct 2019_

### Official Review · AnonReviewer1 · 2019-04-16
**Unclear if using neural nets to approximate lagrangians is a good idea.**

**Rating:** 2
**Confidence:** 2

**Review:**

1. The paper makes a variety of unsupported claims to justify their approach. Mainly this has to do with assuming that reducing the problem to a neural-net parametrized minimax game is inherently a good idea.
   - The paper assumes that GAN-stype minimax problems are easier to solve than having neural nets parametrize the dual. This is a claim that requires substantial elaboration, since it's widely acknowledged that GANs are hard to train, and one would rather have a non-minimax optimization objective than a minimax one.
   - The paper also claims that because neural nets have good inductive biases for tasks such as object recognition, they would serve as good Lagrange multipliers. I'm not sure where this claim comes from. Lagrange multipliers simply need to diverge to indicate constraint violations. These claims about statistical learnability dont seem relevant to the problem.
   - 'direct estimation of probability distribution is not always convenient' seems backwards. Implicit generative models are a necessary evil when you're trying to generate from certain types of neural networks, not that they're more convenient than using MLE.
   - Citing the universal approximation theorem for DNNs is vacuous in this setting considering that you could just solve the original nonparametric problem.. the claim should be that neural nets provide lower approximation error for the amount of required computation.

2. There should be substantially more work analyzing the effect of the Lipschitz constraint on the multipliers.
   - Currently, the Lipschitz constraint comes out of nowhere. If the Lagrange multipliers are unbounded, then you've failed at the minimax objective, and the resulting solution is undefined in the primal. You have an optimal transport map that fails to satisfy the marginal constraints, so it does you no good to add arbitrary restrictions to the Lagrangian.
   - If you want to continue to go down this road - you should really characterize what this Lipschitz constraint does to the objective. Equation (9) is a good start, but its not terribly clear what happens here, because you now have two metrics: one implied by the cost (c) and another one from the Lipschitz constraint. I think you'd be in better shape if you just reformulate the entire paper in terms of W_1 with a fixed distance metric and define the Lipschitz constant there.
   - Finally, I conjecture (I think you might be able to do this by examining the subgradient structure of W_1 near X ...) that if you're in W_1 with eta > 1, then you would actually be solving the original problem. That would be a pretty interesting result, and would actually justify much of the paper.

3. Baselines comparisons seem lacking.
   - A natural approach is just to apply a neural ODE to the pushforward formulation from Monge. Yes, this isn't always feasible, but the proposed SPOT procedure is heuristic to start with. Does this approach not work?
   - The generation experiments don't make sense from the narrative of the paper. If you've actually solved the OT problem, the generated images are trivial (i.e. recovered from the original training set) because you match the marginals. If you're generating new images, then you've failed to solve the OT marginal constraints. Either way this seems problematic.
   - The paper is motivated in terms of continuous OT, but the experiments all operate on discrete, empirical distributions (plus generation experiments, but see my comments above.. the fact that you generalize just means you didn't solve the original OT problem.. this is a negative, not a plus). This is not only a conceptual gap in the paper, it means that the authors should have set up appropriate comparisons to entropically regularized / subsampled OT algorithms which are much more principled and mature.

4. Consider reframing the paper
   - The paper has some interesting results, about generating paired samples, and the importance of doing such tasks and so on. However, the paper is currently motivated as approximating the underlying OT objective. From this latter motivation, the paper is very lacking - there's many conceptual holes about whether the lagrangian constraints are holding, or if this is a good idea, or if you've set up the appropriate baselines.

---

### Official Review · AnonReviewer2 · 2019-04-16
**A clever combination of many ideas in optimal transport and neural density estimation, with surprisingly rigorous experiments**

**Rating:** 5
**Confidence:** 3

**Review:**

This was submitted as a workshop paper but I think with just a little more detail it would be a strong contender for a regular conference paper. The authors show how a combination of primal-dual estimation for Lagrangian problems with neural ODEs for density estimation can be used to solve the regularized optimal transport problem on high dimensional spaces. They show with numerous experiments on challenging domain-to-domain problems that the joint probability distribution they learn can be used for meaningful joint generation tasks, such as pix2pix-like style transfer in the image domain. Overall this paper was a pleasure to read - clearly motivated, tackling an important problem and using state-of-the-art methods to achieve it. I would have appreciated a little more discussion of the stability of gradient ascent/descent for solving the Lagrangian multiplier formulation of their objective, as I have found these kinds of problems very hard to work with in a stochastic domain, but overall the paper was compelling and timely.

---

### Decision · Program_Chairs · 2019-04-19
**Acceptance Decision**

**Decision:**

Accept

**Comment:**

Accepted